# Determinants of promoter and enhancer transcription directionality in metazoans

Mahmoud M. Ibrahim[1,2,5], Aslihan Karabacak[1,2], Alexander Glahs[1,2], Ena Kolundzic [1,2], Antje Hirsekorn[1], Alexa Carda[3], Baris Tursun[1], Robert P. Zinzen [1], Scott A. Lacadie[1,4] & Uwe Ohler [1,2,3,4]

Divergent transcription from promoters and enhancers is pervasive in many species, but it remains unclear if it is a general feature of all eukaryotic cis regulatory elements. To address this, here we define cis regulatory elements in *C. elegans*, *D. melanogaster* and *H. sapiens* and investigate the determinants of their transcription directionality. In all three species, we find that divergent transcription is initiated from two separate core promoter sequences and promoter regions display competition between histone modifications on the $+1$ and $-1$ nucleosomes. In contrast, promoter directionality, sequence composition surrounding promoters, and positional enrichment of chromatin states, are different across species. Integrative models of H3K4me3 levels and core promoter sequence are highly predictive of promoter and enhancer directionality and support two directional classes, skewed and balanced. The relative importance of features to these models are clearly distinct for promoters and enhancers. Differences in regulatory architecture within and between metazoans are therefore abundant, arguing against a unified eukaryotic model.

[1] Berlin Institute for Medical Systems Biology, Max Delbrück Center for Molecular Medicine, 13125 Berlin, Germany. [2] Department of Biology, Humboldt Universitaet zu Berlin, 10115 Berlin, Germany. [3] Department of Biostatistics & Bioinformatics, Duke University Medical Center, Durham 27710 NC, USA. [4] Berlin Institute of Health (BIH), Berlin 10178, Germany. [5] Present address: Department of Nephrology and Immunology, Faculty of Medicine, RWTH Aachen University, Pauwelstraat 30, 52074 Aachen, Germany. These authors contributed equally: Scott A. Lacadie, Uwe Ohler. Correspondence and requests for materials should be addressed to S.A.L. (email: scott.lacadie@mdc-berlin.de) or to U.O. (email: uwe.ohler@mdc-berlin.de)

The application of deep-sequencing assays led to the unanticipated observation that the promoters of many genes are transcribed in both directions, a phenomenon dubbed divergent transcription. In divergent promoters, transcripts made in the direction antisense to the annotated gene are non-protein-coding and highly unstable such that they can typically only be detected in assays enriching for nascent RNA. Divergent transcription is pervasive across many eukaryotes including yeast, *C. elegans*, *M. musculus*, and *H. sapiens*[1–5], though is far less common in *D. melanogaster*[6].

In mammals, the asymmetric output of divergent promoters was suggested to be the result of a post-transcriptional competition model between the splicing machinery and the cleavage/polyadenylation machinery. Here, enriched splice site sequences lead to transcript extension and stabilization in the forward direction, whereas enriched cleavage sequences lead to transcription termination and RNA degradation by the nuclear exosome complex in the reverse unstable direction[7,8]. A different, Nrd1-complex mediated mechanism was found to destabilize divergent promoter transcripts in yeast[5,9,10].

These observations are unable to fully explain transcription directionality since nascent RNA data shows considerable variation in forward/reverse transcription initiation rates[4,6,11]. Divergent promoters initiate transcription from two separate core promoters upstream antisense to each other within a single nucleosome depleted region (NDR), forming two distinct polymerase pre-initiation complexes (PICs)[11–14]. Differences in the sequence-encoded strengths of the forward- and reverse-directed core promoters were reported to potentially drive variation in promoter directionality in *H. sapiens* HeLa cells[11,15], in contrast to recent results obtained using massively parallel reporter assays that measure initiation outside the native genomic context[16,17]. Therefore, asymmetric output of mammalian divergent promoters is potentially sequence-encoded at both transcription initiation and post-transcriptional termination/degradation.

The level of divergent transcription is also reflected in a unique promoter chromatin environment exemplified primarily by differences in levels and distribution of methylation on lysine 4 of histone H3 (H3K4me1/2/3) upstream of the promoter NDR[11,18]. H3K4 methylation and other histone post-translational modifications (PTMs) on promoter NDR-flanking nucleosomes are known to influence transcription initiation and elongation rates via direct physical interactions with PICs[19–21], which may contribute to directional variation of transcription initiation within promoter NDRs.

Divergent transcription is also observed in distal gene regulatory elements such as enhancers, producing and/or long noncoding RNAs with varying stabilities sometimes referred to as enhancer RNAs (eRNAs). Transcriptional activity has been recently identified as a defining feature of active enhancers in mammals[12,22,23]. While enhancers have been long known to feature different chromatin states than those of promoters[24], recent studies have suggested that promoters and enhancers are not distinct types of regulatory elements since they both feature divergent transcription, with H3K4 methylation states varying according to differences in transcription initiation rates[12,25,26]. Of note, the striking similarities in architecture between promoters and enhancers does not necessarily translate to functional equivalence[17,27].

While divergent transcription in mammals is reflected in both DNA sequence and chromatin, the precise contribution of sequence and chromatin features to transcription initiation directionality (i.e., the ratio of forward-to-reverse transcription initiation levels, Fig. 1a) is not well understood. To reconcile seemingly contradictory observations about the prevalence of divergent transcription in different eukaryotes, as well as the mechanisms regulating it, here we quantify the directional relationships between promoter sequence, histone PTMs, and transcription initiation for *Drosophila melanogaster*, *Caenorhabditis elegans*, and *Homo sapiens*. In all three species, we observe strict directional correlations between core promoter sequence strengths and initiation quantity, as well as highly direction-specific correlations between active histone modifications upstream and downstream of promoter NDRs. We find forward/reverse histone modification levels and core promoter sequence strengths alone to be highly predictive of both promoter and enhancer initiation directionality. In our models, sequence and PTMs contribute differently to two regulatory types, directionally balanced transcription initiation and directionally skewed transcriptional initiation, and these contributions differ between enhancers and promoters. Sequence content asymmetry upstream vs. downstream of promoter NDRs is distinct across species and suggests species-specific mechanisms for post-transcriptional contributions to transcript directionality. Finally, low-level divergent transcription initiation is detected from active enhancers in all three species, with putative enhancer activity strongly enriching for divergent transcription in *D. melanogaster* promoters.

## Results

**Variation of promoter initiation directionality**. To identify active promoter and enhancer candidates, we performed the assay for transposase-accessible chromatin (ATAC-seq) on *D. melanogaster* S2 cells and *C. elegans* whole L3-stage to complement previously published data in the *H. sapiens* cell line GM12878[28]. NDRs were then defined using peak-calling with the JAMM algorithm[29], and the resulting peaks were annotated as promoters based on proximity to an annotated gene transcription start sites (TSS, see Methods). This yielded 18,067 promoter NDRs in the *H. sapiens* cell line, 6926 in the *D. melanogaster* cell line, and 10,912 in the L3-stage whole *C. elegans* (Fig. 1a, b).

To assess directionality of transcription initiation for the detected NDRs (Fig. 1a), we used previously published PRO/GRO-cap datasets in *H. sapiens* GM12878, *D. melanogaster* S2 cells and L3-stage whole *C. elegans*[4,12,30]. These nuclear run-on-based assays are sensitive to detecting TSSs of both stable and unstable nascent transcripts from promoters and enhancers. For 14,371 (79%), 6280 (91%), and 10,786 (99%) NDR regions in *H. sapiens*, *D. melanogaster*, and *C. elegans*, respectively, at least one PRO/GRO-cap read mapped to at least one strand. Evaluating the forward (annotated gene) reads against those on the reverse strand enables a minimally biased view of promoter transcription initiation directionality across the three species (Fig. 1c). Beyond basal, likely inactive subpopulations (lower left corners), *H. sapiens* GM12878 cells show some correlation between forward and reverse signal, but with a substantial skew toward the *x*-axis, reflecting promoters with biased directionality toward the annotated gene. Based on other available data, HeLa cells also show bias in directionality toward annotated genes (Supplementary Fig. 1a), consistent with published observations[11]. The skew of initiation toward annotated genes is far greater in *D. melanogaster*, whereas *C. elegans* shows a distribution between *D. melanogaster* and *H. sapiens* (Fig. 1c)[4,6].

To examine whether distinct promoter groups can be discerned based on promoter transcription initiation directionality, we used a Gaussian mixture model to represent forward-to-reverse initiation ratios in promoters that showed sufficient expression in the forward direction (see Methods). Bayesian information criterion analysis of cluster numbers suggested 2, 1, and 2 mixture components as optimal for *H. sapiens*, *D. melanogaster*, and *C. elegans*, respectively (Fig. 1d). Therefore, promoters in *H. sapiens*

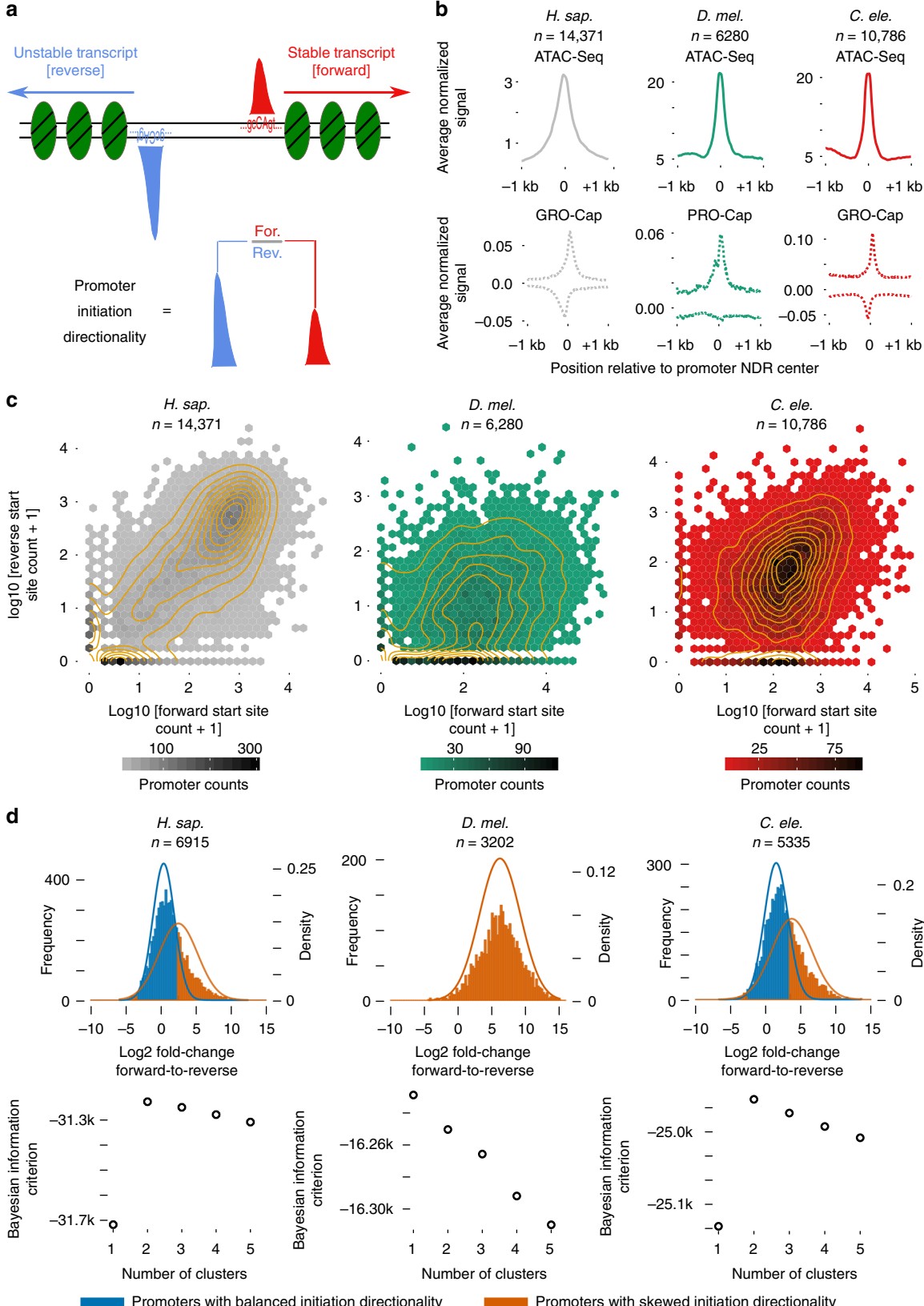

**Fig. 1** Variation of promoter initiation directionality. **a** Schematic of divergent transcription initiation from promoter regions. **b** Average depth-normalized ATAC-seq (solid line) coverage and zero-to-one-scaled PRO/GRO-cap (dotted line) coverage relative to promoter NDR midpoints as defined by ATAC-seq. **c** Forward direction (annotated gene) vs. reverse direction PRO/GRO-cap counts displayed as contour and hexbin scatter plots for the same promoter NDRs as **b**. **d** Mixture model (top) and Bayesian Information Criterion analysis of cluster numbers (bottom) for forward/reverse PRO/GRO-cap count ratios for promoter NDRs containing significant forward initiation. A pseudo count of 1 was added to numerators and denominators. Lines represent density of theoretical Gaussian distributions learned from the data, histograms represent observed ratios

GM12878 and stage L3 *C. elegans* can be grouped into balanced promoters with comparable levels of initiation in both forward and reverse directions and skewed promoters with significantly higher initiation levels in one of the directions, whereas *D. melanogaster* S2 cells supports only one promoter group with initiation levels skewed toward the forward gene (Fig. 1d). Two populations for transcription initiation directionality were also found for *H. sapiens* annotated bidirectional promoters presumably driving stable transcripts in both directions (11% with skewed initiation), further supporting the notion that promoter directionality is not purely determined post-transcriptionally (Supplementary Fig. 1b).

Next, we selected high confidence divergent promoters where both the forward, annotated-gene side, and the reverse, unannotated side of the NDR initiate transcription above a stringent background model (see Methods). While these criteria yielded 5231 and 2670 divergently transcribed promoter NDRs in *H. sapiens* and *C. elegans*, respectively, a substantially smaller number of 441 promoter NDRs met these criteria in *D. melanogaster* S2 cells. To confirm that this *D. melanogaster* set is not simply composed of unannotated stable-stable bidirectional promoters, we generated PEAT (paired-end analysis of TSSs) data in S2 cells[31], an assay which measures TSSs of stable polyadenylated transcripts, and found the reverse signal of the selected divergent promoters to be preferentially depleted in stable transcripts (Supplementary Fig. 1C). Therefore, albeit much less frequent than in *H. sapiens* or *C. elegans*, the selected *D. melanogaster* group is likely to correspond to true unstable-stable divergent promoters, consistent with results from exosome knockdown experiments reported in a concurrent study[32].

**Sequence features of divergent promoters**. The initiation pattern (i.e., the distribution of start site reads across positions within a promoter; Fig. 2a) has been shown to correlate with other promoter properties such as core promoter sequence elements, chromatin modification state, and expression level[33–35]. To assess initiation pattern distributions in divergent promoters, we applied an entropy-based metric[36] to the forward and reverse TSSs within divergent promoters, with lower values indicating high signal over few bases; i.e., so-called focused promoters. Overall, there is a high degree of agreement between forward and reverse initiation patterns in all three species, suggesting that reverse TSSs in our stringent groups are not randomly located events (Supplementary Fig. 1d). Of note, while there is a slight tendency for focused forward promoters to have less relative divergent transcription in *H. sapiens* and *D. melanogaster*, the forward initiation pattern does not seem to be an informative factor for initiation directionality in *C. elegans* (Supplementary Fig. 1e).

To examine the role of the core promoter sequence in directing reverse initiation from divergent promoter NDRs, we turned to position-specific Markov chain models of core promoters[37], which quantify positional enrichment probabilities of sequence k-mers in a test population after being trained on a separate positive training set. As these models score positional sequence composition for a given core promoter sequence and do not utilize previously defined specific motifs, they allow for a consistent framework across species with potentially different motif content. In this way, a low but consistent contribution of multiple individual motifs can be quantified together for a given core promoter. As we previously reported for *H. sapiens* HeLa cells[11], reverse TSSs from GM12878, *D. melanogaster* S2, and *C. elegans* L3 all score well compared to background controls taken from the center of the divergent NDRs (Fig. 2b), which is consistent with the presence of well-positioned TATA and initiator consensus motifs around reverse TSSs (Supplementary Fig. 1f). An

enrichment of known core promoter motifs can also be observed in all three species in the forward direction of forward-skewed promoters compared to balanced promoters (Supplementary Fig. 2a). In all, 1.9%, 1.4%, and 6.1% of divergent promoters have detectable TATA boxes in both forward and reverse core promoters for *H. sapiens*, *D. melanogaster*, and *C. elegans*, respectively, with no clear dependence on the distance between forward and reverse TSSs (Supplementary Fig. 2b). Consistently, a low percentage of divergent promoters have high core promoter sequence model scores (top quartile) in both directions (5% in *H. sapiens*; 0.9% in *D. melanogaster*; 3.8% in *C. elegans*). These data strongly suggest that all three species initiate forward- and reverse-directed transcription from forward- and reverse-directed core promoter sequences within NDRs.

Sequences at forward initiation sites for *D. melanogaster* show substantially higher model scores compared to *H. sapiens* and *C. elegans* core promoters (Fig. 2b), indicating that positional and directional sequences within core promoters are highly prevalent in *D. melanogaster*, which is consistent with previous reports[38,39]. This increased level of well-positioned sequence content in forward core promoters and the large sequence difference between forward and reverse core promoters is very likely to explain the overall scarcity of divergent promoters in *D. melanogaster* (Fig. 1b–d).

Since *D. melanogaster* is known to contain several core promoter motifs not generally found in *H. sapiens* or *C. elegans*, we compared the presence of these motifs surrounding forward and reverse TSSs of divergent promoters as well as forward TSSs of promoters that drive expression only in the forward direction. Consistent with the core promoter model scores discussed above, several motifs are present at reverse TSSs of divergent promoters, but they are substantially less frequent than at forward TSSs of divergent and unidirectional promoters (Supplementary Fig. 3a). Notably, Motif1 is clearly depleted from divergent promoters, as is ChIP-exo signal for Motif1 binding protein (M1BP[40]; Supplementary Fig. 3a, b). Therefore, in addition to a large difference in core promoter strength between forward and reverse TSSs, Motif1 and M1BP may also contribute to the lack of divergent transcription in *D. melanogaster*. For promoters containing the canonical core promoter elements DPE and TATA box, we not only observed a relationship between motif content and core promoter sequence model scores (motif-positivity yields higher scores), but also detected decreased forward model scores for regions harboring core promoter elements in the reverse core promoters (Supplementary Fig. 3c).

**Promoter sequence features are direction-specific**. We next examined the relationships between sequence content and transcription initiation features by means of rank-based partial-correlation analysis, which examines pairwise correlations between two features removing confounding effects due to all the other features. In this manner, we compared ATAC-seq counts, forward and reverse initiation rates as measured by PRO/GRO-cap, initiation pattern entropy scores, and core promoter sequence model scores of divergent promoters in each species (Fig. 2a, c). Strikingly, in all three species forward initiation rate correlates with forward core promoter sequence strength but is independent of reverse core promoter sequence strength. In turn, the reverse initiation rate correlates with reverse core promoter sequence strength but is independent of forward promoter strength. These observations indicate independent initiation events in the forward and reverse directions, and they confirm the key contribution of reverse-directed core promoter sequences to divergent transcription.

Previous research has identified connections between asymmetric sequence content next to divergent promoters and

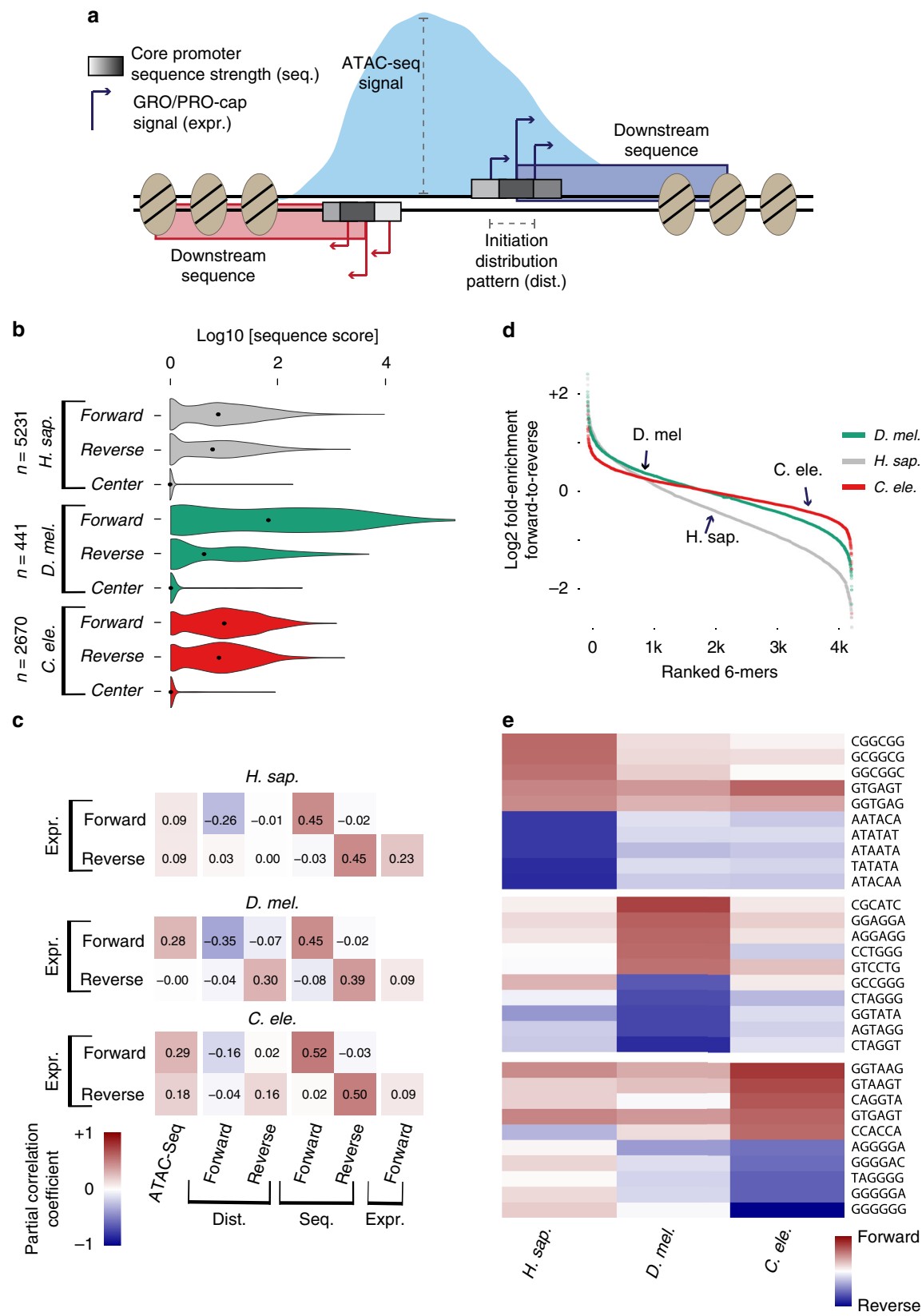

effective transcription elongation in the forward and reverse direction[7,8]. To compare these asymmetries across organisms, we adopted the approach taken by Almada et al.[7] and contrasted the ratio of forward-to reverse six-mer occurrences across 500 bp windows downstream of forward or reverse TSSs (Fig. 2a, d).

Comparing the 5 most forward- and reverse-enriched sequences of each species shows that these asymmetries are unique to each organism, with the 5′ splice site consensus GTGAGT as the only exception (Fig. 2e). For *H. sapiens*, it confirms previous reports of consensus 5′ splice site enrichment in the forward direction and

**Fig. 2** Asymmetric sequence features contribute to variation of transcription directionality. **a** Schematic of features measured for stringently selected divergent promoters. **b** Core promoter sequence model scores at significant forward and reverse TSS modes for promoter NDRs in all three species (see Methods). Center positions between forward and reverse TSSs serve as negative controls. Black dots represent median values. **c** Partial-correlation analysis between total PRO/GRO-cap counts (Expr.), ATAC-seq signal, TSS distribution entropy (Dist.) and core promoter sequence score sums (Seq.) in forward and reverse directions for promoter NDRs with significant forward and reverse TSSs (see Supplementary Table 1 for full partial-correlation table). **d** All 6-mer sequences from 500 bp windows downstream of forward and reverse TSSs for stringently selected divergent promoter NDRs. Sequences are ranked by, and plotted against, their forward-reverse count ratios. **e** Top-5 (red) or bottom-5 (blue) 6-mers (see **d**) from each species with their respective scaled forward/reverse count ratios in the other species

high enrichment of AT-rich, polyadenylation-site-like sequences in the reverse direction. The 5′ splice site is also forward-enriched in *C. elegans* and somewhat less so in *D. melanogaster*, but neither *D. melanogaster* nor *C. elegans* shows enriched AT-rich six-mers in the reverse direction, which is also reflected in the average positional GC content (Supplementary Fig. 3d). The most highly reverse-enriched six-mers in *C. elegans* contain G stretches, leading to a striking pattern of positional GC-skew (Supplementary Fig. 3d). Therefore, sequence asymmetry downstream of forward and reverse TSSs of divergent promoter NDRs is largely distinct across species. Given the known Nrd1-mediated RNA degradation mechanism in yeast[5,9,10], this suggests that the splicing/cleavage competition model of transcript elongation may be limited to vertebrates.

**Promoter chromatin environment is direction-specific**. We took advantage of the high resolution of ATAC-Seq and PRO/GRO-cap assays to study differences between the chromatin organization of *H. sapiens*, *D. melanogaster*, and *C. elegans* promoters. *D. melanogaster* and *C. elegans* promoter NDRs are on average significantly smaller than those in *H. sapiens* (Supplementary Fig. 4a), and their transcription initiation sites are closer to the +1 nucleosome than in *H. sapiens* (Supplementary Fig. 4b). This prompted us to ask whether there are also differences in the spatial arrangement of histone PTMs at promoter regions. We generated ChIP-seq data for H3K4me2 in *D. melanogaster* S2 cells to complement publicly available datasets for H3K4me1, H3K4me3, and H3K27ac in S2 cells[41], as well as all four modifications in *C. elegans* and *H. sapiens*[42,43] (see Methods). We then characterized combinatorial chromatin states for those four PTMs at 10-bp resolution using a multivariate Gaussian Hidden Markov Model (see Methods). We identified 11 chromatin states, which we refer to according to their spatial trends (Fig. 3a): three promoter states (P1, P2, and P3), two transcription elongation states (EL1 and EL2), two enhancer states (E1 and E2), two H3K4me1-only states, one H3K4me2-only state, and a background state (Fig. 3a). Although the discovered chromatin states are similar across species, consistent with previous observations[44], there are differences in their spatial arrangement around promoter NDRs. The forward direction of divergent NDRs in *H. sapiens* and *D. melanogaster* show a cascade of P1-P2-P3 similar to our previous analysis of HeLa cells (Fig. 3b)[11], while the forward direction in the *C. elegans* divergent NDRs is dominated by P2, reflecting a relative confinement of H3K4me2 to the +1 nucleosome in *C. elegans* (Fig. 3b and Supplementary Fig. 4c). An enrichment of the H3K4me1-only state in the center of *C. elegans* NDRs likely reflects the whole-worm aspect of this data, and that some of the selected regions are dynamically accessible across tissues. In the reverse direction, *H. sapiens* and *C. elegans* are enriched for P2 similar to our findings in HeLa cells[11], whereas *D. melanogaster* divergent promoters display an enrichment of P1 on their −1 nucleosome, a state that is enriched only in the forward direction for *H. sapiens* and *C. elegans* (Fig. 3b, compare black lines for P1 and orange lines for P2). Therefore, the histone PTM spatial distribution upstream

and downstream of divergently transcribed promoter NDRs differs across species.

Given that forward and reverse transcription initiation events occur from two independent core promoters (Fig. 2b, c), we wondered whether promoter histone PTM levels upstream of promoter regions are also independent of PTM levels downstream. To test this, we carried out a partial-correlation analysis between the maximum forward and reverse levels of PTM ChIP-Seq signals in a 1 kb window downstream or upstream of promoter NDRs (Fig. 3c). Notably, H3K27ac and H3K4me3 forward (+1 nucleosome) levels show a negative correlation with reverse (−1 nucleosome) levels in all three species (Fig. 3c, follow green squares). When coupled with the expected positive correlation between H3K4me3 and H3K27ac occupying the same nucleosomes (Fig. 3c, follow green squares), this result indicates a strictly direction-specific arrangement of histone PTMs and points to an active competition for histone methyltransferases and acetyltransferases between forward and reverse initiation events in all three species.

**Genuine transcription initiation in active enhancers**. As recent studies have increasingly pointed towards similarities between promoters and enhancers, we next examined transcription initiation at distal enhancer NDRs. We selected ATAC-seq peaks intersecting at least H3K4me1 and H3K27ac and situated far from annotated genes on both strands (i.e., active enhancers candidates[41], see Methods). Forward and reverse nascent transcription initiation levels in those regions indicate that distal active enhancers in all three species display consistent but not necessarily bidirectional transcription initiation (Fig. 4a). In all, 81.8, 89.2, and 99.9% of PTM-selected distal ATAC-seq peaks had at least 1 PRO/GRO-cap read for *H. sapiens*, *D. melanogaster*, and *C. elegans*, respectively, with *C. elegans* enhancers showing the highest level of divergent transcription (Fig. 4a). When selecting enhancer NDRs with transcription above our stringent significance cutoffs (see Methods), we found their core promoter sequences to be of similar strength to those in reverse-directed divergent promoter NDRs (Fig. 4b, compare to Fig. 2b). This indicates that transcription initiation in actively transcribed enhancers is encoded similarly to divergent promoter transcripts in all three species, and that it is not the result of a generic, weak affinity of the polymerase initiation complex to open chromatin.

Gene regulatory elements with enhancer potential can also be identified by high-throughput reporter assays such as STARR-Seq[45], which has been used to collect substantial data for *D. melanogaster*. Intersecting distal ATAC-Seq NDRs with STARR-Seq peaks in S2 cells (i.e., with active enhancer candidates) enriches for divergently transcribed elements relative to distal NDRs that do not intersect STARR-Seq peaks (Fig. 4c and Supplementary Fig. 5a). Additionally, promoter-annotated NDRs that intersect STARR-seq peaks also reveal a strong enrichment for divergent transcription initiation and divergent promoter-like chromatin states compared to promoter NDRs not intersecting STARR-seq peaks (Fig. 4d, e and Supplementary Fig. 5b, c). These observations suggest that (divergent) transcription is indeed a

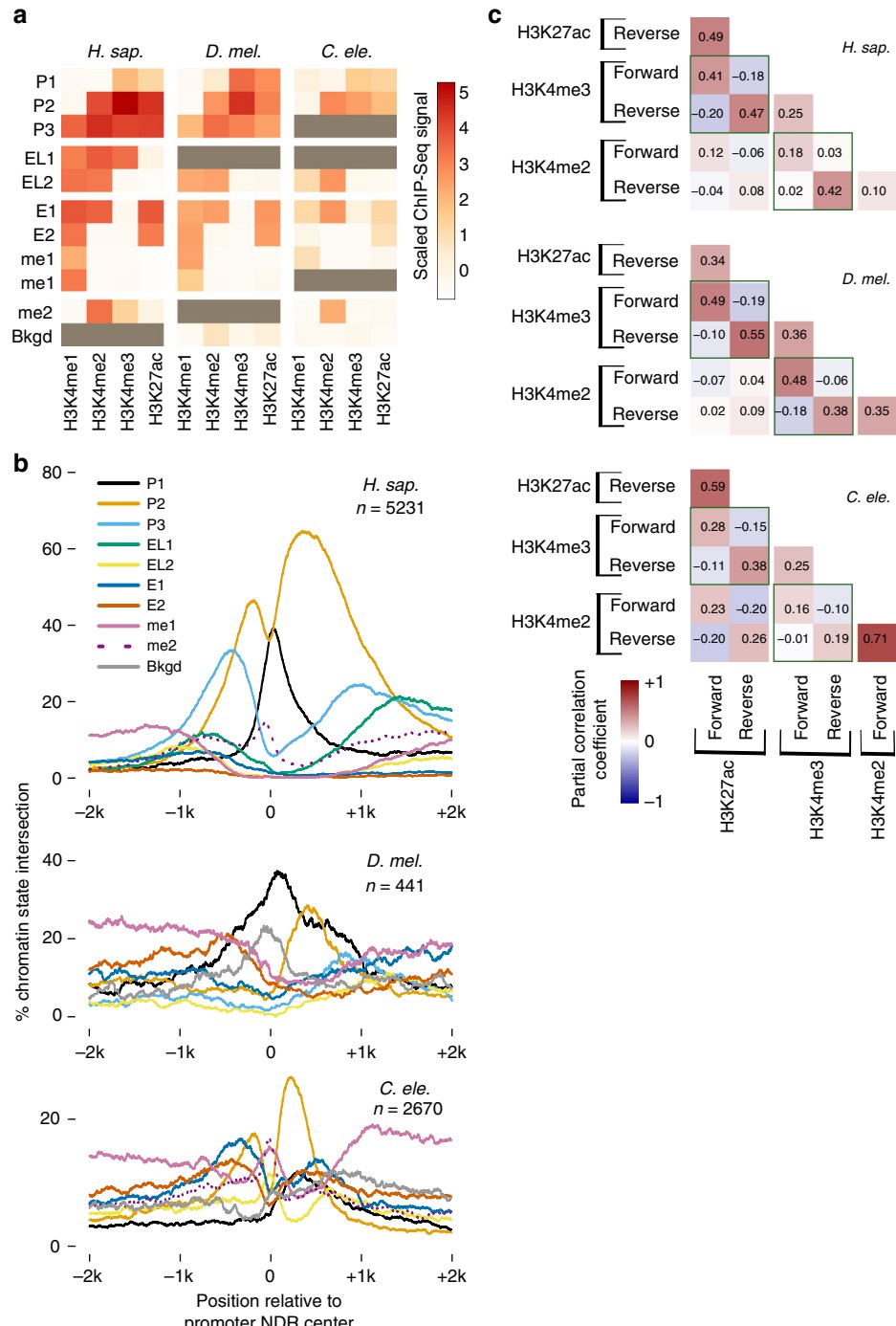

**Fig. 3** Promoter histone PTM states are direction-specific. **a** A heatmap representing chromatin states learned for each species via a multivariate Gaussian Hidden Markov Model. Each state is a multivariate Gaussian distribution and is represented here via the mean scaled ChIP-Seq signal. Gray boxes indicate the state was not detected in the respective organism. P1/2/3: Promoter1/2/3, EL1/2: Transcription elongation1/2, E1/2: Enhancer1/2, me1: H3K4me1, me2: H3K4me2, Bkgd: Background. **b** Chromatin state positional coverage for promoters with stringently selected forward and reverse TSSs. **c** Partial-correlation analysis between promoter histone PTMs ChIP-Seq signal in forward and reverse directions for the same promoter NDRs (see Supplementary Table 2 for full partial-correlation table)

strong indicator for enhancer activity across eukaryotes, consistent with reporter-based activity assays of divergently transcribed *H. sapiens* and *D. melanogaster* enhancers[22,27].

**Variation of enhancer chromatin architecture.** To understand transcription initiation in the context of the chromatin architecture at distal enhancers, we investigated the distribution of

chromatin states (Fig. 3a) at enhancer NDRs. As expected, we found enhancers are on average depleted of promoter-associated states (Fig. 5a) and of H3K4me3 (Supplementary Fig. 6a), but with noticeable variation across species. *H. sapiens* and *C. elegans* enhancers feature weak H3K4me3 enrichment seen in chromatin states P3 and P2 respectively (Fig. 5a), while *D. melanogaster* enhancers have relatively low coverage of E1 (H3K27ac,

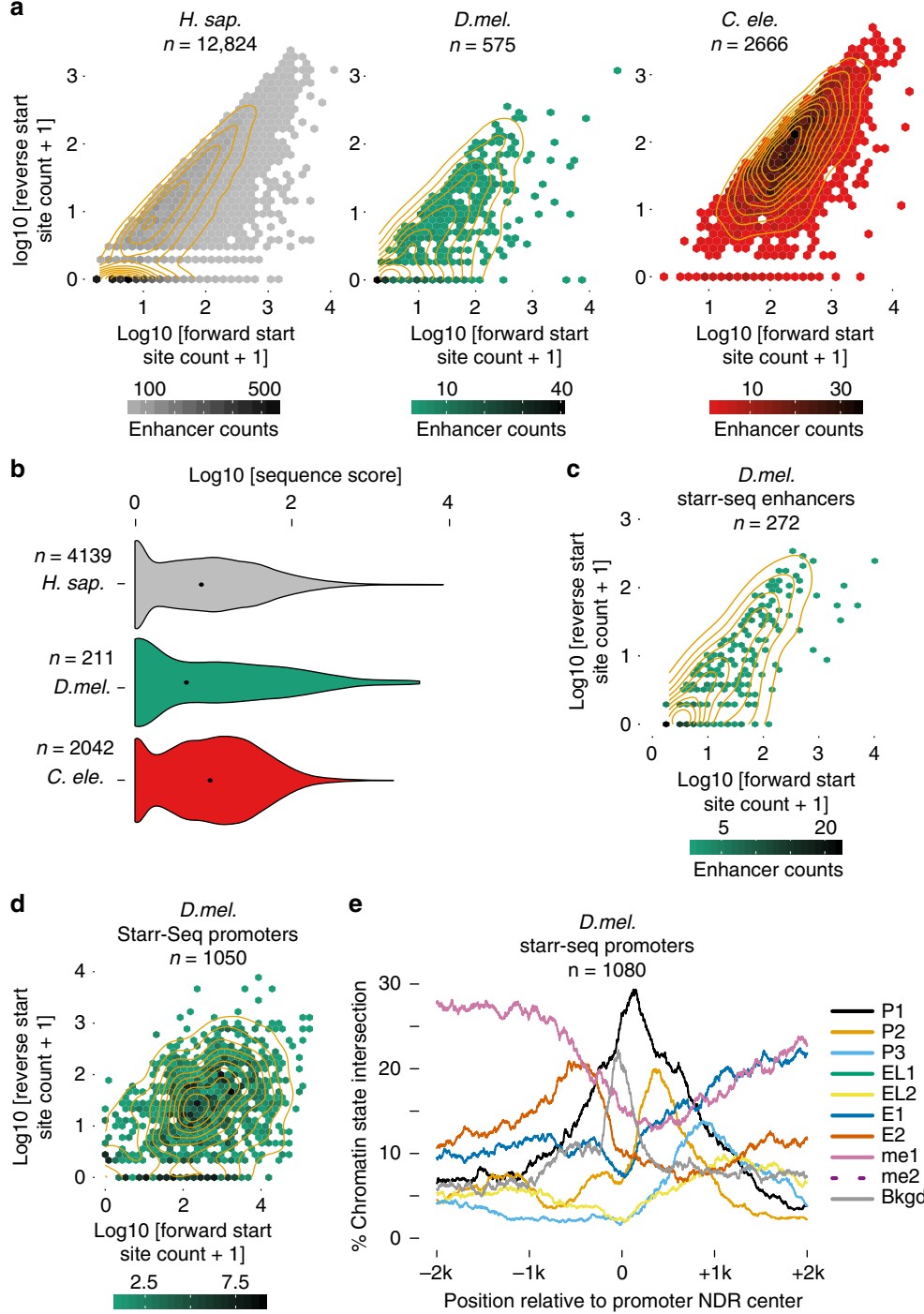

**Fig. 4** Genuine transcription initiation from distal enhancers. **a** Contour/hexbin scatter plots of forward vs. reverse direction PRO/GRO-cap counts for ATAC-seq-defined intergenic NDRs, which intersected at least H3K4me1 and H3K27ac in *H. sapiens* GM12878 (left), *D. melanogaster* S2 cells (middle) and whole L3 *C. elegans* (right). Forward and reverse are the strands with higher and lower counts, respectively. **b** Core promoter sequence model scores at enhancers with significant forward and/or reverse TSS modes (see Methods). Forward and reverse TSSs are plotted together. Numbers indicated in the figure refer to the number of enhancer regions. Total TSSs measured are 5588 (*H. sap.*), 267 (*D. mel.*), and 3062 (*C. ele.*). Black dots represent median values. **c** Forward vs. reverse direction PRO-cap counts for intergenic NDRs that intersect STARR-seq peaks and have at least one count on one strand. **d** Forward vs. reverse direction PRO-cap counts for promoter NDRs that intersect STARR-seq enhancer peaks and have at least one count on one strand. **e** Chromatin state coverage plots for promoter NDRs that intersect STARR-seq enhancer peaks

H3K4me1,me2) and H3K4me2 signal (Fig. 5a and Supplementary Fig. 6a) and still maintain a bimodally-enriched pattern for E2 (H3K27ac, H3K4me1 only). These observations confirm the presence of distal enhancer-like elements in *C. elegans*, which were previously under-characterized and highlighted in a concurrent study[46].

To further investigate the H3K4me3 signal at *H. sapiens* enhancers, we selected active enhancers that intersect H3K4me3

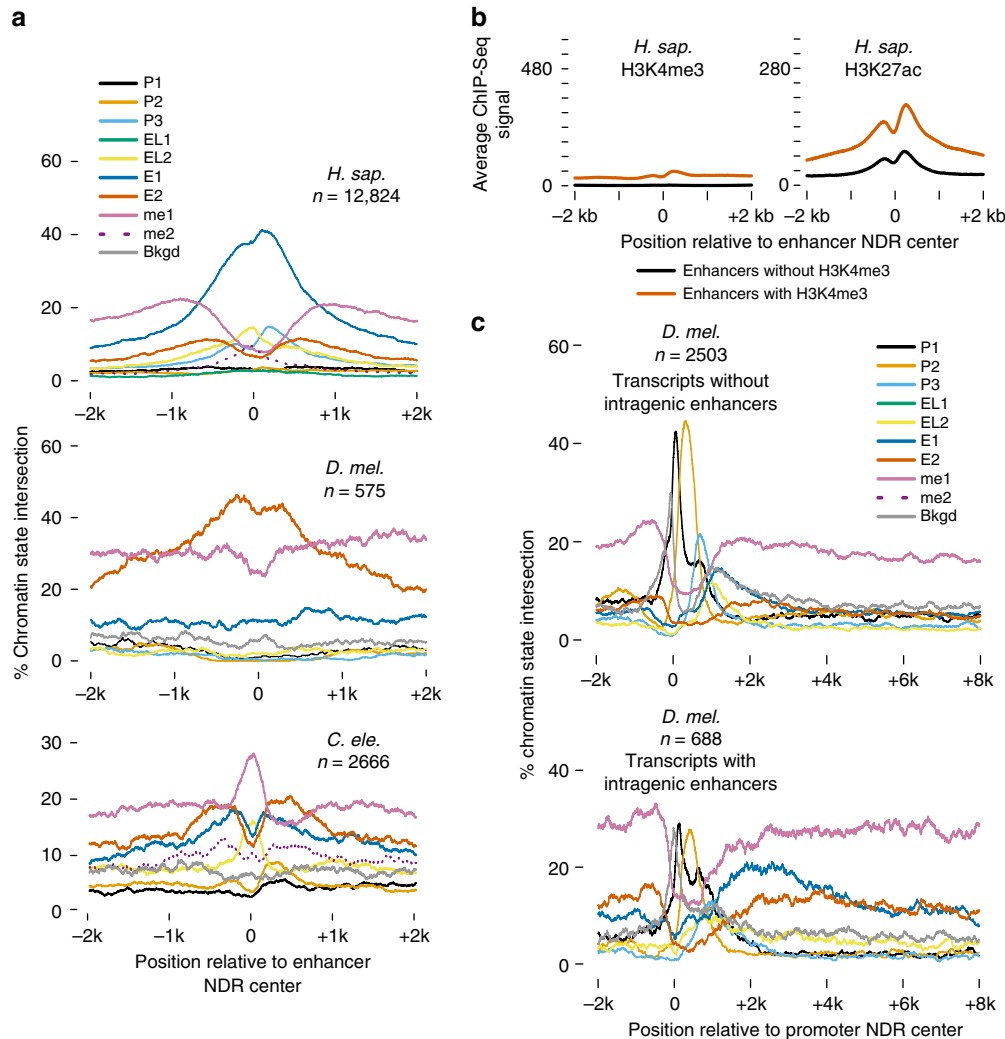

**Fig. 5** Variation of enhancer chromatin architecture across species. **a** Chromatin state (see Fig. 3a) positional coverage for enhancer NDRs (intergenic NDRs that intersect at least H3K4me1 and H3K27ac). **b** Average normalized ChIP-Seq signal for H3K4me3 (left) and H3K27ac (right) at human enhancer regions that either intersect or do not intersect H3K4me3 ($n = 6537$ and $n = 6287$, respectively). **c** Chromatin state positional coverage for expressed promoters whose transcripts are either without (top) or with (bottom) intragenic NDRs that intersect STARR-seq peaks

ChIP-Seq peaks and contrasted them with those that do not intersect H3K4me3 (Fig. 5b). We found H3K4me3 enhancers to feature significantly lower H3K4me3 signal than those of active promoters (Fig. 5b, compare to Supplementary Fig. 4c), but to display higher H3K27ac signal and higher levels of transcription initiation than enhancers without H3K4me3 (Fig. 5b and Supplementary Fig. 6b). Indeed, progressively higher levels of H3K4 methylation correlate with increasing levels of transcription[42] in all three species, although this effect is most pronounced in *H. sapiens* (Supplementary Fig. 6b).

Finally, the absence of bimodally-enriched E1 state (H3K27ac, H3K4me1,me2) in intergenic *D. melanogaster* enhancers (Fig. 5a) prompted us to ask where this chromatin state occurs instead. We classified gene bodies into those that do and do not contain NDRs intersecting enhancer regions as defined by the STARR-seq assay[43]. This leads to a striking enrichment of both E1 (H3K27ac, H3K4me1,me2) and E2 (H3K27ac, H3K4me1 only) states well downstream of the promoter region, but only for the STARR-seq enhancer NDRs (Fig. 5c). Gene Ontology analysis of these genes indicates that they are enriched for functions related to developmental processes (Supplementary Fig. 6c). Altogether with the state pattern in intergenic NDRs (Fig. 5b), these

observations suggest that *D. melanogaster* enhancers tend to have different chromatin architecture depending on whether or not they fall within a gene, and that intragenic H3K4me2-associated enhancers in particular mark developmental and tissue-specific genes.

**A predictive model of initiation directionality**. To quantify the joint impact of the separately characterized sequence and chromatin features associated with transcription initiation directionality, we combined both features in a predictive model of transcription directionality. We chose a mixture model whose components are two separate linear models (see Methods)[47]. This model serves to predict the directionality ratio, with the mixture components representing the directionality type, skewed or balanced initiation (see Fig. 1d). Motivated by the above analyses, we used the ratio of forward-to-reverse core promoter sequence scores and the ratio of forward-to-reverse H3K4me3 PTM levels as predictive features (see Methods).

We trained two separate models for promoters and enhancers and assigned each promoter and enhancer to the type predicted by the model. Examining the distributions of experimentally

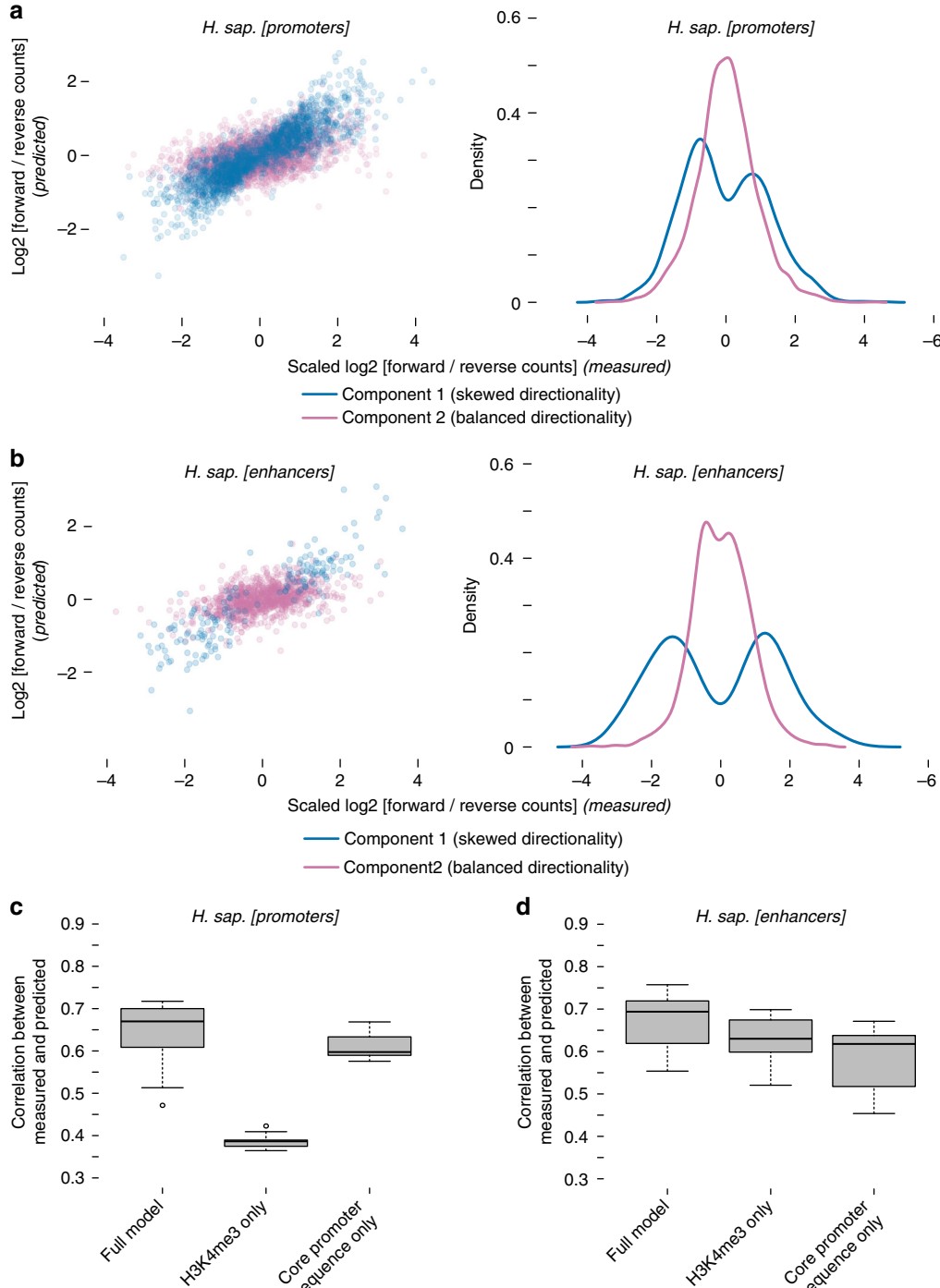

**Fig. 6** H3K4me3 and core promoter sequence are predictive of initiation directionality. Predicted vs. measured transcription directionality in *H. sapiens* **a** promoter and **b** enhancer regions showing significant forward and reverse initiation levels ($n = 5231$ and $n = 1189$, respectively) using a mixture linear model (left). Density plots of **a** promoter and **b** enhancer counts for different directionality groups predicted by the model (right). Models were trained on all promoter (**a**) and all enhancer (**b**) regions. Skewed (blue) and balanced (pink) directionality model components are displayed separately. **c** Tenfold cross-validation of three models including a model trained using core promoter sequence score ratios and H3K4me3 signal ratios, a model trained using core promoter sequence score ratios only and one trained using H3K4me3 ratios only. Boxplots show the correlation between predicted and measured transcription directionality for the test sets. Center line is the median, box represents the 25th and 75th percentiles and whiskers extend further by 1.58 times the interquartile distance divided by the square root of the number of data points. **d** Same as **c** but for enhancers

measured forward/reverse initiation reveals the presence of two groups of promoters/enhancers: one with balanced initiation directionality and one with skewed initiation directionality (Fig. 6a for promoters and Fig. 6b for enhancers). Further, the predicted and experimentally measured forward/reverse transcription

initiation ratios show a correlation of 0.69 for promoters and 0.65 for enhancers. Therefore, forward and reverse core promoter sequences and histone modifications together are highly predictive of transcription initiation directionality levels (Fig. 6a, b). Indeed, using the divergent promoter model to predict

directionality type and level in annotated (stable-stable) bidirectional promoters leads to similar results to that of divergent promoters (Supplementary Fig. 7a), indicating that directionality in bidirectional promoters is also variable and encoded in both sequence and histone modification (Supplementary Figs. 1b and 7a).

We used cross-validation to assess the fit of the full model and to compare it to two models trained on sequence or H3K4me3 levels only. While core promoter sequence scores appeared to be the key important feature for the model's performance in promoters (Fig. 6c), H3K4me3 was the key feature in enhancers (Fig. 6d). Regression coefficients indicate that core promoter sequence scores appear to be more influential for predicting directionality in promoters with skewed directionality than in directionally balanced promoters (Supplementary Fig. 7b), while this relationship is reversed in enhancers (Supplementary Fig. 7c). This analysis lends further support to the hypothesis that promoter directionality is functionally determined by promoter NDR sequence content and adjacent histone PTM levels[11,15,48]. Additionally, it reveals that enhancers, too, feature variable directionality levels encoded in sequence and histone PTM levels. Finally, even though enhancers and promoters have several sequence and chromatin similarities, they might diverge in how those features mechanistically translate into transcription initiation and directionality levels.

## Discussion

We presented a comparative analysis of transcription initiation and histone PTM directionality at promoters and enhancers that highlights several differences between regulatory elements and across species: (1) Promoter initiation directionality is distinct across species with very few divergent promoters in *D. melanogaster* and many in *C. elegans* and *H. sapiens*. (2) There are at least four types of regulatory elements with respect to directionality in *H. sapiens*, skewed and balanced promoters and enhancers. (3) *H. sapiens* enhancers utilize histone PTMs and core promoter sequences differently than promoters to determine initiation directionality. (4) The dual core promoter sequence and chromatin state asymmetry of *D. melanogaster* divergent promoters is clearly different from *C. elegans* and *H. sapiens*. (5) *D. melanogaster* enhancers exhibit two types of chromatin states depending on whether they are intragenic/developmental or intergenic/housekeeping. (6) Asymmetric sequence content surrounding promoters (which determines transcript extension versus termination/degradation in mammals) is clearly distinct across species. Although strong similarities exist, we, therefore, conclude that a single model of transcription initiation within and across eukaryotic species is not evident.

We observe a strict direction-specific correlation between core promoter sequence strength and initiation rate in the forward and reverse directions from promoter NDRs (Fig. 2c). Therefore, forward and reverse-directed transcription events are measurably independent from each other. This is consistent with previous observations of separate pre-initiation complex formation and clear separation of initiation sites[11,12,14,49]. However, it appears not to be consistent with recent reports using massively parallel reporters[16,17], which however lack the native genomic chromatin context. Our analysis also indicates a strict direction-specific positive correlation between different histone PTMs on the +1 and −1 nucleosomes of promoter NDRs in all three species (Fig. 3c). We and others have previously reported a histone PTM arrangement around promoter NDRs reflecting differences in initiation directionality[11,48,49], though RNAPII kinetics and RNAPII PTMs are also likely to contribute[18]. A promoter directionality model emerges whereby direction-specific synergy

between core promoter sequences and histone PTMs in the forward and reverse directions determines fitness in a competition for a common pool of RNAPII[15]. We tested this idea using a linear regression model and found core promoter sequence strength and H3K4me3 levels to be predictive of transcription initiation directionality for both promoters and enhancers (Fig. 6 and Supplementary Fig. 7).

This mixture model also supports the notion of two separate groups for both promoters and enhancers, which we define here as directionally skewed vs. directionally balanced transcription initiation. Those two groups show differences in how the synergy between sequence and chromatin is coordinated: while promoters with skewed directionality are mainly determined by core promoter sequence, histone PTMs play a bigger role for predicting directionality of balanced promoters. Interestingly, the contribution of these features to enhancer initiation directionality is different (Fig. 6 and Supplementary Fig. 7), pointing to a distinction between promoters and enhancers that, to our knowledge, has not been previously described.

We propose a refined picture of transcriptional directionality in which (a) skewed directionality is enforced at genuine endogenous promoters, where one side acquired functionality to transcribe a functional trans-acting (m)RNA at relatively higher levels, consistent with recent studies by Jin et al.[50]; (b) transcription of a divergent product (functional or not) may also act as a tuning mechanism for the initiation rates of a functional, oppositely-oriented counterpart; (c) the directional variation of initiation across NDRs is determined by directionally competing sequence and chromatin features; and (d) apparently species-specific mechanisms ensure that any divergent, nonfunctional transcripts are efficiently degraded.

Our finding that enhancer activity overlapping annotated promoter regions in *D. melanogaster* S2 cells enriches for divergent transcription (Fig. 4) has been verified with embryonic transgenesis in a concurrent study[27], has also been shown in *H. sapiens* cells[51], and is consistent with previous observations using the CAGE assay in mammalian cells[22]. These observations potentially point to a mechanistic function for divergent transcription at enhancers. Transcription initiation may help to position nucleosomes, thereby ensuring accessibility to the DNA by transcription factors, and nascent eRNA may act to compete with chromatin for nucleic acid binding factors/complexes[52]. On the other hand, enhancers have a different functional requirement than promoters: they do not need to produce stable transcripts at possibly high levels, as exemplified by recent studies[17,27]. It is, therefore, possible that enhancers with skewed directionality act in a mechanistically distinct way, e.g., as promoters for lncRNAs which subsequently act *in trans* as transcriptional regulators, but such distinctions remain to be addressed. As the nascent transcriptomes of more eukaryotes are profiled, we anticipate that a wide range of transcription directionality tendencies will be observed with different chromatin-sequence synergy mechanisms.

## Methods

*C. elegans* **ATAC-seq**. *C. elegans* wild-type strain N2 was grown on OP50 bacteria at 20 °C as described before (Brenner, 1974). Embryos were harvested from adults by sodium hypochlorite treatment and grown until third larval instar (L3). Synchronized L3 animals were washed five times in M9 buffer and collected on ice. Nuclei were isolated using a glass Dounce homogenizer with 50 strokes tight-fitting insert in buffer A (15 mM Tris–HCl pH7.5, 2 mM MgCl2, 340 mM sucrose, 0.2 mM spermine, 0.5 mM spermidine, 0.5 mM phenylmethanesulfonate [PMSF], 1 mM DTT, 0.1% Trition X100 and 0.25% NP-40 substitute) as described before[53,54]. The debris were removed by spinning at $100 \times g$ for 5 min and nuclei were counted by Methylene blue staining. In all, 100.000 nuclei per sample were pelleted by spinning at $1000 \times g$ for 10 min and proceeded immediately to transposition step of the ATAC-seq protocol[28]. Briefly, nuclei were incubated in 1x tagmentation buffer (TD buffer Illumina Nextera Kit, FC-121-1030) supplemented with 2.5 microlitres

of Tn5 transposon (TDE1, Illumina Nextera Kit, FC-121-1030) for 60 min at 37 °C. DNA was purified using the DNA Clean and Concentrator-5 kit (Zymo Research), followed by PCR amplification using 1x High-fidelity NebNext PCR master mix (New England Biolabs). Libraries were amplified for a total of 13 or 14 cycles.

**D. melanogaster S2 cell ChIP-seq, ATAC-seq, and PEAT.** For ChIP-seq, _D. melanogaster_ S2 cells were obtained from Life Tech (#R69007) and grown at 25 °C in Schneider's Cell medium (Life Tech, #21720024) with 10% FBS (Sigma, #F7524) and 10% L-Glutamine (Sigma, #G7513) without antibiotics. Cells were grown in T75 flasks at 25 °C to a confluency of ~ 70%. For ATAC-seq, cells were grown at 25 °C in ExpressFive SFM medium (Life Tech, #10486025) with 10% heat-inactivated FBS (Life Tech, #16000044) and 12% L-Glutamine (Life Tech, #25030024) and 1% Penicillin-Streptomycin (Life Tech, #15070063). Cells were grown in dishes to a confluency of ~ 80–95%.

For H3K4me2 ChIP-seq, formaldehyde was added to media to a final concentration of 1% and incubated for 10 min on a shaker at room temperature. The reaction was quenched by adding glycine to a final concentration of 125 mM followed by 5 min of incubation on a shaker at room temperature. The cells were collected by centrifugation at 500 x g for 5 min at 4 °C and washed twice with ice-cold PBS. The cell pellet was resuspended in 10 ml ice-cold cell lysis buffer (5 mM HEPES (pH 8), 85 mM KCl, 0.5% NP-40) with protease inhibitors (cOmplete™ ULTRA Tablets, Mini, EDTA-free, EASYpack Protease Inhibitor Cocktail, Roche # 05892791001) and incubated for 10 min at 4 °C. Nuclei were released by 10 strokes with a Wheaton Dounce Homogenizer (tight pestle). The crude nuclear extract was collected by centrifugation at 500 x g for 5 min at 4 °C, resuspended in 1 ml ice-cold nuclear lysis buffer (50 mM HEPES (pH 8), 10 mM EDTA, 0.5% N-Lauroylsarcosine with protease inhibitors) and incubated for 20 min at 4 °C. After addition of 1 ml nuclear lysis buffer samples were sonicated using a Diagenode Bioruptor for 18 cycles (30" ON / 30" OFF) on high. After sonication samples were centrifuged at 14000 x g for 10 min at 4 °C and the supernatant was aliquoted to DNA-low binding tubes. The chromatin was flash frozen in liquid nitrogen and stored at −80 °C.

Protein A Sepharose (PAS) beads (Sigma #P9424) were washed twice with RIPA140 (140 mM NaCl, 10 mM Tris–HCl (pH 8), 1 mM EDTA, 1% Triton X100, 0.1% SDS, 0.1% Na-Deoxycholate) with proteinase inhibitors and 1 mg/ml BSA (Sigma #A7906) and incubated overnight. Chromatin was thawed on ice. Fifty micrograms of Chromatin was used per ChIP experiment. RIPA140 with proteinase inhibitor was added to a total volume of 1 ml and incubated with 2 µg of H3K4me2 antibody (Abcam, ab32356, Lot#: GR209821-1; or Epicypher 13-0013, Lot#: 14247001) for 16 h at 4 °C on a rotating mixer (40 rpm). One percent of the chromatin was used as input controls. Blocked beads were added to the chromatin–antibody complex solution and incubated for 3 h at 4 °C on a rotating mixer at 40 rpm. Complexes were washed once with 1 ml RIPA140, 4 times with 1 ml RIPA500 (500 mM NaCl, 10 mM Tris–HCl (pH 8), 1 mM EDTA, 1% Triton X100, 0.1% SDS, 0.1% Na-Deoxycholate) for 10 min each. Complexes were subsequently washed once in 1 ml LiCl-Buffer (250 mM LiCl, 10 mM Tris–HCl (pH 8), 1 mM EDTA, 0.5% NP-40, 0.5% Na-Deoxycholate) and TE (10 mM Tris–HCl (pH 8), 1 mM EDTA) for 2 min each. Between each wash, beads were spun down at 500 x g for 2 min and the supernatant was discarded. Beads were resuspended in 100 µl TE and RNase A was added to a final concentration of 50 µg/ml followed by incubation at 37 °C for 30 min. The samples were adjusted to a final concentration of 0.5% SDS and Proteinase K was added to a final concentration of 500 µg/ml. Proteins were digested at 37 °C for 90 min followed by reverse cross-linking overnight at 65 °C. DNA was purified using phenol–chloroform extraction followed by ethanol precipitation. Libraries were prepared using the NEXTflex qRNA-Seq Kit v2 from Bioo Scientific (Catalog #5130-11) and sequenced on an Illumina NextSeq 500 sequencer.

For ATAC-seq, 200,000 cells were subjected to tagmentation as described above. Libraries were amplified for a total of 13 or 15 cycles.

For PEAT data, _D. melanogaster_ S2 cells were grown to a density of 3 million per ml in Schneider's Medium (Invitrogen) supplemented with 10% FCS and 1x Antibiotics. The PEAT library was constructed as described[31].

**Previously published datasets.** ChIP-seq datasets for H3K4me1, H3K4me2, H3K4me3, and H3K27ac from _C. elegans_ whole l3 stage were downloaded from data.modencode.org[43] corresponding to experiment IDs 5048, 5157, 3576, and 5054, respectively. ChIP-seq datasets for H3K4me1, H3K4me3, and H3K27ac from _D. melanogaster_ S2 cells were downloaded from the Gene Expression Omnibus (GEO; Series GSE41440[41];). ENCODE ChIP-seq datasets for H3K4me1, H3K4me2, H3K4me3, and H3K27ac from _H. sapiens_ GM12878 cells were downloaded corresponding to GEO sample IDs GSM733772, GSM733769, GSM945188, and GSM733771, respectively, as well as input control sample IDs GSM733742 and GSM945259.

GRO-cap datasets for _H. sapiens_ GM12878 cells (Series GSE60456[12];) and _C. elegans_ whole L3 stage (Series GSE43087[4];) were downloaded from GEO.

ATAC-seq datasets for _H. sapiens_ GM12878 cells were downloaded from GEO (Series GSE47753[28];).

M1BP ChIP-exo was downloaded from GEO (Series GSE97841[40];)

**Data processing.** PRO/GRO-cap datasets were subjected to adapter removal using cutadapt[55] as was ATAC-seq using flexbar[56] prior to mapping. Reads were then mapped with Bowtie2[57] with default settings, including the parameter –X 1500 for paired-end datasets, to the hg19, ce6, or dm6 genome assemblies, followed by removal of multi-mapped reads from the resulting.sam files. ChIP-Seq datasets were aligned using bowtie2 with default parameters and reads that had more than 2 mismatches and did not align uniquely were removed. M1BP ChIP-exo was mapped with bowtie2 (-X 1500,—no-mixed). The sequencing library for S2 H3K4me2 ChIP-Seq dataset was prepared with Unique Molecular Identifiers, therefore the first 9 bases of each read were removed using flexbarv2.4[56] and reads were aligned to dm6 genome build using bowtie2 in paired-end mode keeping only concordantly aligned mates. All ChIP-seq datasets were collapsed using samtools rmdup[58] and ATAC-seq and M1BP ChIP-exo datasets were collapsed using MarkDuplicates.jar from Picard tools (http://broadinstitute.github.io/picard). Duplicates were not removed from PRO/GRO-cap datasets. ATAC-seq read pairs with fragments greater than 50 bp were kept for further processing. Start sites of ATAC-seq reads were extended by 15 basepairs upstream and 22 basepairs downstream in a stranded manner, to account for steric hindrance of the trans-position reaction[59]. All reads that intersected ENCODE blacklisted regions (https://sites.google.com/site/anshulkundaje/projects/blacklists) were removed and all replicate BED files were concatenated together for peak-calling and signal gen-eration. Signal bigwig files for ATAC-Seq were generated using JAMM signal generator pipeline[29]. Depth-normalized, read extended bigwig file for M1BP ChIP-exo was generated and plotted using deeptools[60].

PEAT data was processed as follows. Fastq files from each mate were first matched and trimmed for the 5′-end adapter using cutadapt[55] (parameters -a GTTGGACTCGAGCGTACATCGTTAGAAGCT -O 30 -m 20—untrimmed-output). The sequences that were not matched for 5′-end were then matched and trimmed for the 3′-end adapter using cutadapt[55] (parameters -a GTCGGATAGGCCGTCTTCAGCCGCCTCAAG -O 30 -m 20—untrimmed-output). The two resulting fastq files matching each end were combined, reverse complemented, and the unpaired mates discarded and paired mates matched based on read IDs using custom scripts. The resulting paired fastq files were then mapped using STAR[61] (parameters—outFilterMultimapNmax 1—outFilterMismatchNmax 1—chimSegmentMin 30—chimJunctionOverhangMin 30—outFilterIntronMotifs RemoveNoncanonicalUnannotated—alignIntronMin 20—alignIntronMax 1,000,000—alignEndsType EndToEnd—alignMatesGapMax 1,000,000—alignSJoverhangMin 12—alignSJDBoverhangMin 3) to the dm6 genome assembly.

**ATAC-seq peak-calling, annotation, and selection.** ATAC-seq peaks were called using JAMM v1.0.7rev5 setting the bin size to 100, mode (-m) to normal and resolution (-r) to peak for all species. To obtain appropriate peak signal cutoffs each dataset, we used an automated threshold cutoff on window enrichment by setting –e to auto for the gm12878 dataset and used the all list output from JAMM. For the _C. elegans_ L3 and _D. melanogaster_ S2 datasets, no window enrichment cutoff was used (-e 1) but we used the filtered output which performs filtering on the peak level. Only peaks that are larger than 50 basepairs were kept for all species. To ensure that PRO/GRO-cap start site counts in promoter regions are not under-estimated due to narrow-width peak calls, final peaks were extended by 75 base-pairs in each direction and overlapping peaks were merged. Extended, merged, ATAC-seq peaks were annotated as promoters if they were within $+/-$ 200 bp from Gencode defined transcript starts for _H. sapiens_, $+/-$ 400 bp (larger due to compact genome to better filter bidirectional promoters) from Flybase defined transcript starts for _D. melanogaster_, and $+/-$ 500 bp (larger distances to coun-teract inaccuracies in TSS annotations due to trans splicing and due to a compact genome to better filter bidirectional promoters) from refGene defined gene starts for _C. elegans_. Peaks that fell within these distances on both strands were regarded as bidirectional promoters and removed from the analysis. Peaks were annotated as intergenic if they were not annotated as a promoter and did not intersect known transcript boundaries from the same databases.

ATAC-peaks containing confident transcription start sites from nascent RNA datasets were selected based on empirical distributions of read 5′-ends per base from control regions. Control regions were selected as follows: first all ATAC-seq peaks containing at least one nascent RNA read 5′-end on at least one strand were selected, then windows equal in size to a given ATAC-seq peak were taken immediately downstream of that ATAC-seq peak on both strands (i.e., higher coordinates on the plus strand and lower coordinates on the minus strand), and any overlapping regions with other peaks were subtracted out. These windows were taken to represent frequently observed signal background within gene bodies that is likely to come from technical issues in the PRO/GRO-cap protocols (i.e., non-nascent RNA contamination or inefficient cap-selection) and, therefore, are unlikely to represent true transcription start sites. The empirical distribution of nascent RNA read 5′-ends per base was constructed across all the bases in the control windows and cutoffs were determined using the 0.999 quantile for _C. elegans_ (17 read 5′-ends) and _D. melanogaster_ (8 read 5′-ends), and the 0.9999 quantile for _H. sapiens_ (14 read 5′-ends). Thus, we had two sets of ATAC-seq peaks for downstream analysis: Set A contained at least one base with these numbers of reads on the forward strand only and Set B contained at least one base with these numbers of reads on both the forward and reverse strands. Set B peaks were filtered

for convergent TSS pairs defined when the bases on both strands containing the most read 5′-ends are both over the determined cutoff and are situated downstream from each other.

Non-extended peaks called using JAMM setting –m to narrow to obtain more accurate peak edges were used for promoter peak width and distance between TSS and peak edge analysis (Supplementary Fig. S3A, B). Distance of TSS to the ATAC-Seq peak edge was determined using TSSs defined via extended, merged ATAC-Seq peaks (see above) and narrow non-extended, non-merged peak edges. This allowed for TSSs occurring outside, downstream of the peak (negative distances) and inside the peak (positive distances). Peak width and distance from TSS to peak edge were done using the forward TSS of both Set A and Set B promoter peaks (see above). If the absolute TSS distance to the edge was larger than 200 bp, it was not included in the violinplot in Fig. S3B.

HeLa-S3 DNase-I peaks and 5′GRO-seq data used in Fig. S1A were obtained from Duttke et al. 2015b[11].

Average positional depth-normalized (ATAC-seq) or zero-to-one normalized (PRO/GRO-cap) counts were plotted using ggplot2[62].

**Histone modification peak-calling and signal files.** All histone modification peaks were called using JAMM v1.0.7rev5 setting the bin size to 150 and –r to window[29]. For S2 H3K4me2 dataset, peaks were called using JAMM v1.0.7rev5 in paired-end mode, since this was the only ChIP-Seq paired-end dataset used in this study. The filtered output peaks produced by JAMM was used for all histone modification datasets.

Histone modification bigwig signal tracks were generated using deepTools[60] bamCoverage at 10 bp resolution using the fragment length obtained by JAMM and setting normalization to RPKM. To generate average meta-plots, deepTools computeMatrix was used at single basepair resolution. The bigwig files were also used to define the features for partial correlations and the predictive linear model.

**TSS sequence model.** The TSS sequence model initially described by Frith and colleagues[37] was used as described previously[11]. Set A ATAC-Seq peaks (see above) were used for training species-specific first-order (di-nucleotide) models as follows: a window + /− 50 bp surrounding the position with the most nascent RNA read 5′-end counts within the ATAC-seq peak was used to train the TSS sequence model. The model was then run either on the same corresponding windows from both strands of the selected promoters (Set B ATAC-Seq peaks, Fig. 2b; see above) or NDRs exceeding our stringent cutoff in distal ATAC-seq peaks (Fig. 4b). Midpoints between forward and reverse TSSs served as negative controls for sequence model scores.

Alternatively, the model was run-on windows surrounding all bases on each strand that had at least one nascent RNA read 5′-end and the scores added per strand. The subsequent values were divided by the ATAC-seq peak width and used for the partial-correlation analysis and/or directionality linear mixture modeling (Figs. 2c, 6, and Supplementary Figs. S6; see below). This was carried out on annotated TSS-proximal Set B peaks (see above) for promoters, and intergenic Set B peaks further selected to intersect ChIP-seq peaks for H3K4me1, H3K4me2, and H3K27ac for enhancers.

**TSS initiation pattern score.** Distribution pattern (DP) scores were calculated based on the equation for Shannon entropy similar to a previously described method[36]. Specifically,

$$DP = - \sum_{i}^{n} p_i \log_2 p_i \qquad (1)$$

where $p$ is the probability of a nascent RNA read 5′-end at position $i$ for a given strand of an ATAC-seq peak and $n$ is the total number of all the positions for that strand that have at least one read 5′-end.

**GC content and skew.** GC percentage was calculated in a sliding 50 bp window with a step size of 1 bp along a given region and then taking the positional mean across all selected regions. GC-skew was calculated as follows:

$$GC_{skew} = \frac{(g - c)}{(g + c)} \qquad (2)$$

where $g$ and $c$ are the number of G and C nucleotides in a 50 bp window slid along a given region with a step size of 1 bp. Positional means were then calculated across all regions and plotted with ggplot2[62].

**Motif scanning.** Motifs were scanned using SpeakerScan[63] with a zero-order Markov background window of 100 bp. Scores below zero were set to zero, and scores were plotted with ggplot2[62] either as heatmaps (Fig. S2A) with maximum color set to the 0.999 quantile of all scores and scores above that set to maximum color, scatter plots (Fig. S2B) where maximum scores were taken between position −45 and −25 of the TSS for the TATA motif, or positional averages (Fig. S3). For violin plots in Fig. S3C, core promoters were considered positive if they had a score

greater than zero between positions −45 and −25 of the TSS for the TATA motif or between positions + 20 and + 30 for the DPE motif.

**Mixture modeling.** Mixture models of ATAC-seq peak nascent RNA read 5′-end count ratios were calculated using the R package Mclust with default parameters[64].

**Chromatin State Hidden Markov Model.** Histone modification peaks were processed for the chromatin state HMM as previously described[11]. Briefly, histone modification peak calls were used to threshold the data by obtaining ChIP-Seq signal where there are peaks and zeros elsewhere. Regions without peaks in all analyzed histone modifications are discarded. ChIP-Seq read counts are log-scaled to obtain a distribution resembling a Gaussian distribution. Chromatin states were then obtained also as previously described[11] at 10 basepair resolution using multivariate Gaussian distribution for the emission probabilities, but with the following changes: only sequences that were at least 500 basepairs long were kept for Baum-Welch training, which was performed while tying the transition probability matrix to 0.9 at the diagonal and to $0.1/(n − 1)$ at all other entries (where $n$ is the number of states); the segmentation was obtained using posterior decoding. Baum-Welch was run-on chromosome 1 for gm12878 and on all chromosomes for S2 and L3. The two H3K4me1-only states were added and plotted as one line in state coverage plots in all figures.

**Partial-correlation analysis.** Histone modification features were defined as the maximum histone modification ChIP-Seq signal in a 1 kb window downstream and upstream of the non-extended ATAC-seq peaks (see above) for forward and reverse directions, respectively. The initiation rate was calculated as the total number of PRO/GRO-cap read 5′-ends on each respective strand of the extended ATAC-Seq peaks divided by ATAC-Seq peak width. As core promoter sequence feature, we added all model scores for positions with at least one PRO/GRO-cap read 5′-end and divided by ATAC-Seq peak width (see above). ATAC-Seq features were defined as the maximum ATAC-Seq signal in the non-extended ATAC-seq peaks.

Spearman partial-correlation coefficients were then obtained using the R package ppcor[65] and heatmaps were plotted using the R package pheatmap[66].

**Transcription directionality model.** Features were defined as for the partial-correlation analysis (see above). A linear mixture model was learned using the R package flexmix[47], setting the number of clusters to 2. All features were standardized to have a mean of zero and standard deviation of 1 before training the model. H. sapiens promoter and enhancer regions were selected to have divergent TSSs that meet the stringent count cutoff for forward and reverse initiation sites (Set B; see above) and enhancers were further selected to intersect H3K4me1, H3K4me2, and H3K27ac peaks (see above). This results in learning two linear models with distinct regression coefficients and assigning each data point a probability of belonging to each of the two mixture components. Each promoter region is then assigned to the mixture component that has the higher probability. To obtain predicted directionality ratios, the prediction from the mixture component that the promoter region is assigned to is used.

For cross-validation, the data was split into ten equal parts and model learning and clustering were repeated ten times; each time the model predictive ability was tested on the held-out test set, summarized using the correlation coefficient. This was done separately for three different models, one that included both core promoter sequence score ratio and H3K4me3 ratio and two that included sequence score ratio only or H3K4me3 only.

***D. melanogaster* enhancer analysis.** STARR-Seq[45] peaks were obtained from the Stark lab website (http://www.starklab.org/data/arnold_science_2013/) and coordinates were lifted over to dm6 genome assembly. Peaks from both replicates were merged and extended by 200 bp in each direction. For Fig. 5c, promoters that belonged both Set A ATAC-Seq peaks and Set B peaks were chosen and stratified by whether their corresponding transcript intersected a STARR-Seq peak that intersected an ATAC-Seq peak. For Fig. 5d, all promoter annotating ATAC-Seq peaks were stratified by whether they intersect a STARR-Seq peak.

***D. melanogaster* gene ontology analysis.** Gene Ontology Biological Process enrichment was determined using the PANTHER database overrepresentation test (release: 20171205) and the Panther GO-Slim Annotation set (version 13.1, released 03-02-2018)[67–70]. GO terms were filtered for those that are overrepresented with FDR value smaller than 0.01 and whose reference set contains at least 100 genes but no more than 2000 genes.

**Code availability.** HMM-related scripts for Baum-Welch training and posterior decoding are available at https://github.com/mahmoudibrahim/hmmForChromatin. All other custom code is available upon request.

## Data availability

The data that support the findings of this study are available from the corresponding author upon request. *D. melanogaster* S2 PEAT, *D. melanogaster* S2 ATAC-seq, L3-stage *C. elegans* ATAC-Seq data and *D. melanogaster* S2 H3K4me2 ChIP-Seq data are available at the Gene Expression Omnibus under accession number GSE103177. Access information regarding previously published data is detailed in the Methods section.

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

## Acknowledgements

We would like to thank Hans-Hermann Wessels for help with S2 cell culture as well as James T. Kadonaga, Sascha Duttke, Sven Heinz, and Chris Benner for helpful comments and discussions. M.M.I. was funded by the MDC/NYU exchange program. S.A.L is a BIH Delbrueck fellow. U.O. and A.C. acknowledge ARRA supplement funds for award NIH R01HG004065.

## Author contributions

S.A.L., M.M.I. and U.O. designed the project; S.A.L and M.M.I. performed the analysis with assistance from A.K. and advice from U.O. S.A.L, M.M.I and U.O. wrote the manuscript. A.K. and A.H. performed the ATAC-seq in *D. melanogaster* S2 cells. E.K. performed the ATAC-seq in *C. elegans* with advice from S.A.L., B.T. and A.H. A.G. performed the *D. melanogaster* S2 cell H3K4me2 ChIP-seq with advice from R.Z. and A.H. prepared the library for sequencing. A.C. performed the *D. melanogaster* S2 cell PEAT.

## Additional information

**Competing interests:** The authors declare no competing interests.

