## [peer review file · Nature Communications]

Reviewer #2 (Remarks to the Author):

The revised manuscript by Ibrahim et al shows some improvement, although the improvements in clarity of the text and the responses to all referees have also laid bare some problems that were originally obscured by presentation. It is still a highly valuable study with lots of interesting data and novel observations.

Major comment

The main problem that I have with the way the data is analysed is that using measures such as model scores and partial correlation fails to provide clear answers to simple and obvious questions, such as e.g. - how strong is the antisense initiation in promoters containing strong TATA box at the proper position compared to promoters that clearly do not? How common are the promoters that have strong TATA and/or DPE in both directions. It seems that DPE promoters NEVER have DPE at +25 in other direction - the DPE signal overlapping TSS is an artefact of the fact that DPE shares part of its motif with the initiator - so the question is - what motifs initiate antisense initiation in DPE promoters where it is present? In the apparent antisense initiation that has TATA at -30, what does the forward promoter contain? Is there a relationship between motif content and spacing between forward and reverse initiation? I understand that a paper such as this cannot answer each of these questions explicitly, but some advanced visualisation (such as heatmap-like visualisation of the relationship between PRO/GRO-cap signal and motif content) would reveal more than the current analyses could ever hope to. Jumping to summary scores too early obscures the properties of individual promoters by averaging out key dependencies between forward and reverse sequences.

Minor comments

The writing is still impenetrable at places, and could benefit from editing by somebody with an "ear" for clumsy style. An example of such writing is a sentence from the abstract: "Where present, all three species initiate divergent transcription from separate core promoter sequences and show directional independence between histone modifications." Here "where present" looks like "where all three species are present" instead of the intended meaning, which is probably "Where

transcriptional divergence is present, it is initiated from separate core promoter sequences and shows directional independence between histone modifications in all three species."

Also, the paper (and the rebuttal) read like the authors have developed their own jargon in the process of working on this paper, and now write like this jargon is general knowledge.

Abstract: "positional enrichment of chromatin states is variable across species" - "variable" is misleading here, "different" is better.

The result sections are of very unequal length - the first one is seven pages long, the other three a page each. I suggest breaking up the first section into sensible units.

Page 6, line 122: The reference to (Figure 1A) actually seems to refer to Figure 1B.

Reviewer #2 (Remarks to the Author):

The revised manuscript by Ibrahim et al shows some improvement, although the improvements in clarity of the text and the responses to all referees have also laid bare some problems that were originally obscured by presentation. It is still a highly valuable study with lots of interesting data and novel observations.

We thank the reviewer for their continued interest in our manuscript. We believe addressing the reviewers' previous and current comments, as explained below, has improved the paper significantly.

We have updated our manuscript and included new analyses (new Figure S2 and Figure S3). Key new edits to the manuscript text are highlighted in yellow.

Additionally, we also provide analysis here in the response letter which was not included in the manuscript due to space or to avoid confusing the readers with too many figures. Throughout the response letter, we refer to these analyses as "Response Figure".

Major comment

The main problem that I have with the way the data is analysed is that using measures such as model scores and partial correlation fails to provide clear answers to simple and obvious questions,

We understand the reviewer's point-of-view and concern regarding the lack of a clear answer as to what determines promoter directionality in the context of specific core promoter motifs. We have updated our manuscript to better explain our reasoning for using primarily model scores rather than motif scans. We also further elaborate here our point-of-view regarding why our analysis does not focus on core promoter motifs. We mention two main aspects:

1- In Figures 6 and S7, we provide a predictive quantitative model of transcription initiation directionality. We believe this is the first attempt at such a quantitative model and is, in our opinion, the best way to address the central question of this

manuscript: “What determines transcription initiation directionality in a given nucleosome-depleted region?”. Our answer: the features we used in this model (core promoter sequence strength, perhaps regardless of a specific motif content and histone modification levels), to the extents indicated by our cross-validation results (Figure 6C). Sequence model scores allowed us to move elegantly to this quantitative model of transcription initiation directionality.

2- We had turned to sequence model scores and correlation analyses after having observed, during our work on this manuscript as well as previous manuscripts and unpublished analyses (Duttke et al., Mol. Cell 2015 and Lacadie et al. FEBS 2016), that no single motif would likely explain the role of core promoter sequence in promoter directionality. The main point we strive to emphasize in this current manuscript is that directionality differs greatly between and within organisms, and that the imbalance between forward and reverse core promoter strengths, which we approximate via model scores, is a major determinant of initiation directionality. The sequence model allowed us to apply the same framework to each species regardless of general differences in motif content. As the reviewer indicated in previous comments, increased core promoter model scores reflect increased faithful and well positioned core promoter motifs and vice versa. In order to further emphasize this point, we have included new heatmaps in Figure S2A showing that increased TATA and Initiator motifs in the forward direction correlate with increased forward-to-reverse initiation ratio. We also show the relationship between strong positional motifs and sequence model scores for TATA and DPE in *D. melanogaster* (Figure S3C).

such as e.g. - how strong is the antisense initiation in promoters containing strong TATA box at the proper position compared to promoters that clearly do not?

Overall, our analysis shows that increased reverse initiation reflects increased reverse-directed positional core promoter sequence content AND/OR decreased forward-directed positional core promoter sequence content. In *C. elegans* and *H. sapiens*, this sequence content is likely to be largely reflecting TATA- and initiator-like motifs as suggested by our metaplots of string matches (Figure S1F). To more clearly demonstrate this, we have included new heatmaps in Figure S2A showing how forward core promoter Initiator and TATA motif strength increases with

increasing skew of initiation in the forward direction. The inverse is also true for reverse core promoters as shown in the Response Figure A, especially for C. elegans and H. sapiens.

How common are the promoters that have strong TATA and/or DPE in both directions. It seems that DPE promoters NEVER have DPE at +25 in other direction - the DPE signal overlapping TSS is an artefact of the fact that DPE shares part of its motif with the initiator - so the question is - what motifs initiate antisense initiation in DPE promoters where it is present?

In the apparent antisense initiation that has TATA at -30, what does the forward promoter contain?

Is there a relationship between motif content and spacing between forward and reverse initiation?

I understand that a paper such as this cannot answer each of these questions explicitly, but some advanced visualisation (such as heatmap-like visualisation of the relationship between PRO/GRO-cap signal and motif content) would reveal more than the current analyses could ever hope to. Jumping to summary scores too early obscures the properties of individual promoters by averaging out key dependencies between forward and reverse sequences.

We thank the reviewer for these points. We have chosen to address them collectively with two analysis types. In Figure S2B, we have now included scatter plots for each species of well-positioned TATA motif scores in the forward direction against the reverse direction with each dot color-coded by the distance between the forward and reverse initiation sites. These plots give the reader a sense of how frequently the motifs occur on both sides and suggest little dependence on distance between divergent TSSs. We have included the same type of plots for combinations of TATA and DPE in D. melanogaster in the Response Figure B as well as numbers in the table below based on liberal log-likelihood cutoffs. It should be noted that, though metaplots of motif scores show very little well-positioned DPE signal for reverse core promoters, there are some that show low, but positive, log-likelihood signal as displayed in Response Figure B, but the overall distribution of reverse-directed DPE scores is far below that for the forward direction. Indeed, this highlights a strength of the sequence model we use throughout the manuscript to

detect overall low levels of well-positioned sequence content. We have not included these DPE analyses in the manuscript because they are necessarily *D. melanogaster*-specific, and, therefore, we feel they are details appreciated by a narrow audience of aficionados, going beyond the goal of our study to directly compare divergent transcription between species.

In contrast, forward and reverse TATA motifs can be addressed and compared in all three species and seem to co-occur to slightly different extents in different species ($97/5231 = 1.9\%$, $6/441 = 1.4\%$, and $163/2670 = 6.1\%$, for *H. sapiens*, *D. melanogaster*, and *C. elegans*, respectively). These numbers have been added to the text and complement the new scatter plots in Figure S2B which are further colored by distances between divergent TSSs. These numbers also complement our previous statement in the manuscript that “a low percentage of divergent promoters have high scoring core promoter sequences (top quartile) in both directions (5% in *H. sapiens*; 0.9% in *D. melanogaster*; 3.8% in *C. elegans*)”.

	TATA for	TATA rev	DPE for	DPE rev
TATA for	55	6	14	20
TATA rev	6	50	18	9
DPE for	14	18	150	42
DPE rev	20	9	42	115

In Figure S3C, we have included a plot directly linking positional motif content via scanning with sequence model scores for *D. melanogaster* DPE and TATA motifs. This plot is discussed in the manuscript upon introduction of the sequence model as confirmation of its ability to capture such information, together with a new brief explanation of our reasoning for using the sequence model.

Overall, we feel we have chosen several key analyses that address as many of the issues raised by the reviewer as possible without expanding the figures too much and maintaining the clear goal of the manuscript to compare transcription directionality across species. We hope the reviewer agrees with our efforts to find this balance.

Minor comments

The writing is still impenetrable at places, and could benefit from editing by somebody with an "ear" for clumsy style. An example of such writing is a sentence from the abstract:

"Where present, all three species initiate divergent transcription from separate core promoter sequences and show directional independence between histone modifications."

Here "where present" looks like "where all three species are present" instead of the intended meaning, which is probably "Where transcriptional divergence is present, it is initiated from separate core promoter sequences and shows directional independence between histone modifications in all three species."

We thank the reviewer for pointing this out, we have carefully edited the manuscript and hopefully improved the overall readability.

Also, the paper (and the rebuttal) read like the authors have developed their own jargon in the process of working on this paper, and now write like this jargon is general knowledge.

We have strived to revise this in the new revised text whenever we detected it. Also, to make the paper more accessible, we added a new graphic (new Figure 1A) to serve as a visual aid to the introduction of the terms we use throughout the paper.

Abstract: "positional enrichment of chromatin states is variable across species" - "variable" is misleading here, "different" is better.

We have replaced “variable” with “different” in the indicated sentence.

The result sections are of very unequal length - the first one is seven pages long, the other three a page each. I suggest breaking up the first section into sensible units.

Thank you for pointing this out. We have followed the reviewer’s recommendation.

We believe this section is now significantly more accessible to readers.

Page 6, line 122: The reference to (Figure 1A) actually seems to refer to Figure 1B.

We revised all Figure references, including this one, also after adding a new Figure S2 and new graphic Figure 1A.

Reviewer #2 (Remarks to the Author):

The revised manuscript by Ibrahim et al shows some improvement, although the improvements in clarity of the text and the responses to all referees have also laid bare some problems that were originally obscured by presentation. It is still a highly valuable study with lots of interesting data and novel observations.

Major comment

The main problem that I have with the way the data is analysed is that using measures such as model scores and partial correlation fails to provide clear answers to simple and obvious questions, such as e.g. - how strong is the antisense initiation in promoters containing strong TATA box at the proper position compared to promoters that clearly do not? How common are the promoters that have strong TATA and/or DPE in both directions. It seems that DPE promoters NEVER have DPE at +25 in other direction - the DPE signal overlapping TSS is an artefact of the fact that DPE shares part of its motif with the initiator - so the question is - what motifs initiate antisense initiation in DPE promoters where it is present? In the apparent antisense initiation that has TATA at -30, what does the forward promoter contain? Is there a relationship between motif content and spacing between forward and reverse initiation? I understand that a paper such as this cannot answer each of these questions explicitly, but some advanced visualisation (such as heatmap-like visualisation of the relationship between PRO/GRO-cap signal and motif content) would reveal more than the current analyses could ever hope to. Jumping to summary scores too early obscures the properties of individual promoters by averaging out key dependencies between forward and reverse sequences.

Minor comments

The writing is still impenetrable at places, and could benefit from editing by somebody with an "ear" for clumsy style. An example of such writing is a sentence from the abstract: "Where present, all three species initiate divergent transcription from separate core promoter sequences and show directional independence between histone modifications." Here "where present" looks like "where all three species are present" instead of the intended meaning, which is probably "Where transcriptional divergence is present, it is initiated from separate core promoter sequences and shows directional independence between histone modifications in all three species."

Also, the paper (and the rebuttal) read like the authors have developed their own jargon in the process of working on this paper, and now write like this jargon is general knowledge.

Abstract: "positional enrichment of chromatin states is variable across species" - "variable" is misleading here, "different" is better.

The result sections are of very unequal length - the first one is seven pages long, the other three a page each. I suggest breaking up the first section into sensible units.

Page 6, line 122: The reference to (Figure 1A) actually seems to refer to Figure 1B.

REVIEWERS' COMMENTS:

Reviewer #2 (Remarks to the Author):

The authors have adequately addressed my original concerns.

We would like to thank the reviewer for a very fair and constructive review process.